# The Impact of Heat Acclimation on Gastrointestinal Function following Endurance Exercise in a Hot Environment

**DOI:** 10.3390/nu15010216

**Published:** 2023-01-01

**Authors:** Daichi Sumi, Haruna Nagatsuka, Kaori Matsuo, Kazunobu Okazaki, Kazushige Goto

**Affiliations:** 1Research Center for Urban Health and Sports, Osaka Metropolitan University, Osaka 558-8585, Japan; 2Research Fellow of Japan Society for the Promotion of Science, Tokyo 102-0083, Japan; 3Graduate School of Sports and Health Science, Ritsumeikan University, Shiga 525-8577, Japan; 4Department of Health & Sports Science, Kawasaki University of Medical Welfare, Okayama 701-0193, Japan

**Keywords:** endurance training, heat acclimation, gastrointestinal damage, gastric emptying rate

## Abstract

To determine the effects of heat acclimation on gastrointestinal (GI) damage and the gastric emptying (GE) rate following endurance exercise in a hot environment. Fifteen healthy men were divided into two groups: endurance training in hot (HOT, 35 °C, *n* = 8) or cool (COOL, 18 °C, *n* = 7) environment. All subjects completed 10 days of endurance training (eight sessions of 60 min continuous exercise at 50% of the maximal oxygen uptake (${\overset{\cdot }{V}O}_{2\max}$). Subjects completed a heat stress exercise tests (HST, 60 min exercise at 60% ${\overset{\cdot }{V}O}_{2\max}$) to evaluate the plasma intestinal fatty acid-binding protein (I-FABP) level and the GE rate following endurance exercise in a hot environment (35 °C) before (pre-HST) and after (post-HST) the training period. We assessed the GE rate using the ^13^C-sodium acetate breath test. The core temperature during post-HST exercise decreased significantly in the HOT group compared to the pre-HST (*p* = 0.004) but not in the COOL group. Both the HOT and COOL groups showed exercise-induced plasma I-FABP elevations in the pre-HST (*p* = 0.002). Both groups had significantly attenuated exercise-induced I-FABP elevation in the post-HST. However, the reduction of exercise-induced I-FABP elevation was not different significantly between both groups. GE rate following HST did not change between pre- and post-HST in both groups, with no significant difference between two groups in the post-HST. Ten days of endurance training in a hot environment improved thermoregulation, whereas exercise-induced GI damage and delay of GE rate were not further attenuated compared with training in a cool environment.

## 1. Introduction

The gastrointestinal (GI) tract digests and absorbs nutrients and serves as a barrier to bacterial translocation. Exercise causes GI damage and dysfunction of gut barrier, evidenced by reduced small intestinal permeability, increased bacterial translocation, and inflammation [1,2,3,4]. Elevated levels of exercise-induced plasma intestinal fatty acid binding protein (I-FABP) are an indirect marker of damage to the small intestine [5]. van Wijck et al. [2] demonstrated that resistance exercise increases I-FABP levels and delays the appearance of ingested amino acids in blood. Moreover, exercise-induced plasma I-FABP elevation negatively correlated with digestion and absorption rate following amino-acid ingestion [2]. Thus, exercise-induced GI damage is associated with reduced GI function (slower digestion and absorption of nutrients). Although several factors exacerbate exercise-induced GI damage, reduction of GI blood flow may be involved [2,5,6]. Since blood flows principally distributed to working muscles (to deliver oxygen) and the skin (to dissipate heat) during exercise, GI blood flow thus decreases, triggering intestinal cell hypoperfusion and ischemia that cause hypoxia and oxidative stress, increasing cell damage [6,7]. Furthermore, low GI blood flow delays gastric emptying (GE) [8], which is important in terms of nutrient digestion in stomach [9]. Therefore, the GE rate has often been used to evaluate GI function during and after exercise [10,11,12].

Exercise in a hot environment causes greater GI damage than does exercise in a thermoneutral environment [13,14,15]. Since exercise in a hot environment increases core temperature compared with exercise in a neutral environment, GI hypoperfusion has been suggested to be exacerbated due to marked increases in skin blood flow and sweat rate to dissipate heat [8,16,17]. On the other hand, several days of endurance training in a hot environment facilitates heat acclimation (HA), leading to increase in baseline plasma volume, lower rectal and skin temperature during endurance exercise in a hot environment [18,19]. These adaptations following HA would cause maintenance of GI blood flow during endurance exercise. Thus, we hypothesized that HA would attenuate exercise-induced GI damage and the delay in the GE rate in a hot environment. However, the effect of HA on exercise-induced GI damage and dysfunction remains unclear. It is necessary to investigate whether exercise-induced blood I-FABP elevation after heat acclimation attenuate or not.

As mentioned above, GI damage and dysfunction delay the appearance of ingested macronutrients (glucose and amino acids) in blood [2], which may compromise post-exercise recovery by slowing muscle glycogen and muscle protein synthesis. Short-term endurance training in a hot environment (HA training) improves thermoregulatory capacity and endurance capacity [20,21,22]. More data on post-exercise GI damage/function and post-exercise recovery are required; endurance athletes commonly include several days of HA training in their schedules. Here, the purpose of the present study was to determine the effects of HA on GI damage and the GE rate after endurance exercise in a hot or a cool environment.

## 2. Materials and Methods

### 2.1. Subjects

The present study is placed as a follow-up determination of previous research which reported the effect of HA on the hepcidin response following endurance exercise in a hot environment [23]. Thus, the subjects and experimental design were thus identical to that study [23]. For determination of sample size, we referred to previous studies that demonstrated large effects of thermoregulatory adaptations (e.g., lower core temperature during endurance exercise in a hot environment) after HA [24,25]. Consequently, fifteen healthy males participated. Inclusion of criteria were healthy active male and non-smoking. The mean (standard deviation [SD]) age, height, body mass, fat-free mass, and fat mass were 23 ± 2 years, 170.8 ± 7.2 cm, 66.6 ± 9.4 kg, 55.4 ± 6.8 kg, and 11.0 ± 3.4 kg, respectively. All subjects were physically active, involved in recreational resistance or endurance exercises. All subjects were informed of all experimental procedures and the possible risks; all gave written informed consent. The present study was approved by the Ethics Committee for Human Experiments at Ritsumeikan University, Japan (no. BKC-IRB-2021-014).

### 2.2. Experimental Design

Fifteen healthy men were divided into two groups: endurance training in a hot (HOT, 35 °C, 50% relative humidity (RH), *n* = 8) or a cool environment (COOL, 18 °C, 50% RH, *n* = 7). All subjects completed 10 days of endurance training (eight sessions in total), each of 60 min continuous exercise at 50% of the maximal oxygen uptake (${\overset{\cdot }{V}O}_{2\max}$). As shown in Figure 1, all subjects underwent a heat stress exercise test (HST) before and after completion of endurance training to evaluate the thermoregulatory response during exercise in heat environment (35 °C, 50% RH). Plasma I-FABP levels and the ^13^C-labeled carbon dioxide output (the ^13^CO_2_/^12^CO_2_ ratio in expired gas) were determined after each HST to evaluate changes in exercise-induced GI damage and the GE rate. All training sessions and HST were conducted during December to January to avoid potential effect of seasonal heat acclimation.

### 2.3. Heat Stress Test (HST)

Before training, and on the day after completion of training, subjects came to the laboratory at 09:00 following an overnight fast and rested for 20 min before baseline blood sampling and rectal temperature (T_rec_) measurement. Then, all subjects completed the HST. During each HST, T_rec_, mean skin temperature (T_sk_), heart rate (HR), thermal sensation (TS), and rate of perceived exertion (RPE) were continuously monitored. Before and immediately after each test, body weights were measured to calculate total sweat loss [23]. After test completion, each subject rested in a chair for 1 h in thermoneutral environment (23 °C, 50% RH). Before and immediately after completing the exercise, blood samples were taken from the antecubital vein. Breath samples (for determination of the ^13^CO_2_/^12^CO_2_ ratio) were taken every 5 min during the 60 min post-exercise period.

### 2.4. Endurance Training

All subjects conducted endurance training using a cycle ergometer (Aerobike 75XLIII; Konami Co., Kanagawa, Japan) in an environmentally controlled chamber. To prevent fatigue, days 6 and 7 of the training period were rest days (Figure 1). During each training session, all subjects performed 60 min endurance exercises at 50% of ${\overset{\cdot }{V}O}_{2\max}$ in HOT or COOL environment. Data on T_rec_, HR, TS, RPE, and sweat loss were collected during training on the first day (day 2) and final day (day 11).

### 2.5. Measurements

#### 2.5.1. Maximal Oxygen Uptake (${\overset{\cdot }{V}O}_{2\max}$)

The ${\overset{\cdot }{V}O}_{2\max}$ test was conducted in a cool environment. The test began 1.0 kilopond (kp) and this was increased in 0.5 kp increments every 2 min to exhaustion (at 60 rpm). During the test, expired gases were collected and analyzed through an automatic gas analyzer (AE300S, Minato Medical Science Co. Ltd., Japan). The respiratory data were averaged every 30 s.

#### 2.5.2. HST measurements

##### Thermoregulatory Variables

Details procedures of the T_sk_ and T_rec_ measurements were reported in our previous study [23]. The T_sk_ was measured at the chest, arm, thigh, and calf using probe thermometers (NK543 Nikkiso Co., Kanagawa, Japan); the mean skin temperature was calculated as follows following Ramanathan et al. [26]: T_sk_ = 0.3 × (chest + arm) + 0.2 × (thigh + calf).

T_rec_ was measured using a wired rectal thermistor (ITP010-11; Nikkiso Therm Co. Ltd., Tokyo, Japan) inserted to 10 cm beyond the anal sphincter. T_rec_ and T_sk_ were monitored at 0.5 Hz throughout the experiments using a data logger (N543; Nikkiso Therm Co. Ltd., Tokyo, Japan). The increase in T_rec_ (ΔT_rec_) during the HST as an index of bodily heat storage was calculated by subtracting T_rec_ at rest from the T_rec_ peak during HST. The HR (Accurex Plus; Polar Electro Oy, Kempele, Finland) was measured every 1 min during the HST, and the average value during the 59–60 min was determined. The subjects scored their RPE [27] and TS [28] at the end of each HST. The sweat loss during the HST was calculated using the changes in body weight before and after exercise [23].

##### Blood Variables

After overnight fasting, all subjects visited the laboratory at 09:00 and rested for 20 min before blood collection. A polyethylene catheter was inserted into an antecubital vein and a baseline blood sample was obtained. Blood samples were collected before and immediately after completing the HST. Blood samples for determination of blood hemoglobin [Hb] and the hematocrit [Hct] levels were collected using 2.5 mL syringes containing heparin; we calculated changes in plasma volume (PV). The PV was used to correct the plasma variables [29]. A 10 mL syringe was used to obtain plasma samples that were subjected to 10 min of centrifugation at 4 °C (3000 rpm); all samples were stored at –80 °C prior to analysis. Plasma I-FABP levels were measured in duplicate and averaged using an enzyme-linked immunosorbent assay (ELISA) kit (R&D Systems, USA); the intra-assay coefficient of variation was 2.5%. Blood Hb and Hct levels were measured using an automatic blood-gas analyzer (OPTI CCA TS, Sysmex Co., Japan). All analyses were completed within 15 min after blood collection; all samples were put on ice prior to analysis. The changes in baseline PV after endurance training were calculated using the equation of Dill and Costill [29]:ΔPV (%) = 100 × ([Hbpre/Hbpost] × [100–Hctpost]/[100–Hctpre]–1).
where Hct is the hematocrit level and Hb is the hemoglobin level.

##### Gastric Emptying Rate

The GE rate was evaluated using the ^13^C -sodium acetate breath test [30,31,32]. All subjects ingested 100 mg 1-^13^C -sodium acetate (99 atom %; Cambridge Isotope Laboratories Inc., Andover, Massachusetts, USA) dissolved in 100 mL purified water immediately after completing the HST. Before consumption, a baseline breath sample was collected into a 1.3 L bag (Otsuka Pharmaceutical Co., Ltd., Tokyo, Japan). Twelve breath samples were collected every 5 min during the 60 min post-exercise period; the ^13^CO_2_ levels were determined and the ^13^CO_2_/^12^CO_2_ ratios were calculated using an infrared spectrometer (POC One, Otsuka Pharmaceutical Co., Tokyo, Japan). Changes in the ^13^CO_2_/^12^CO_2_ ratio are expressed as the absolute increases between samples obtained during exercise and at baseline. The ^13^CO_2_ and^12^CO_2_ abundance ratios were converted into the real amounts of excreted ^13^C and employed to evaluate 13C kinetics. The CO_2_ production level per unit of body surface area was assumed to be 300 mmol/m2/h. Body surface area was estimated using the formula of Du Bois and Du Bois [33]. The times at which ^13^C -excretion/h were maximal (i.e., T_max_ values) were closely correlated with scintigraphically evaluated GE rates [30,31,32].

#### 2.5.3. Measurements during Training (Days 2 and 11)

The T_rec_ and HR were measured every 1 min during exercise, and the averages during the 59–60 min of 60 min training were determined. RPE and TS data were evaluated at the end of the 60 min. Total sweat loss was calculated by the reduction in body weight after exercise [23].

### 2.6. Statistical Analyses

Data are expressed as means ± SD. The two-way (one between-group and one within the training period) analysis of variance (ANOVA) repeated measures approach was used to detect significant effects of exercise on all variables in both groups. Similarly, three-way (one between-group and two within the training period and time) ANOVA was used to always detect significant effects of exercise on blood variables during the HST in both groups. When ANOVA revealed a significant interaction or main effect, the Tukey-Kramer post hoc test was performed to identify the differences. For all tests, a *p* value < 0.05 was considered to indicate statistical significance.

## 3. Results

### 3.1. ${\overset{\cdot }{V}O}_{2\max}$ and the Pedaling Workload

The ${\overset{\cdot }{V}O}_{2\max}$ before training (COOL 47.7 ± 4.0 mL/kg/min, HOT 49.3 ± 8.0 mL/kg/min), pedaling workloads during training (COOL 1.7 ± 0.2 kp, HOT 1.8 ± 0.4 kp), and HST (COOL 2.2 ± 0.2 kp, HOT 2.4 ± 0.5 kp) did not significantly differ between the two groups.

### 3.2. Thermoregulatory and Subjective during Training

Table 1 shows the peak T_rec_, HR, sweat loss, TS, and RPE during training; these data were presented previously [23]. As expected, the peak T_rec_ (*p* = 0.046), HR (*p* = 0.003), sweat loss (*p* = 0.008), TS (*p* < 0.001), and RPE (*p* = 0.007) were significantly higher in the HOT than the COOL group throughout training. The HOT group presented declines in the peak T_rec_ (*p* = 0.031), HR (*p* = 0.001), TS (*p* = 0.013), and RPE (*p* = 0.005) on day 11 compared to day 2; we found no significant interaction between training period and group for each variable.

### 3.3. Thermoregulatory and Subjective during the HST

Table 2 shows the changes in T_rec_ (Rest and peak), ΔT_rec_, peak T_sk_, the HR, and sweat loss during the HST (previously reported by Sumi et al. 2022). The rest T_rec_ was not significantly different after training in both groups (*p* = 0.073); we found no significant main effect of group (*p* = 0.707) or any interaction between training period and group (*p* = 0.434). After training, peak T_rec_ (*p* = 0.004) and ΔT_rec_ (*p* = 0.040) were significantly decreased in the HOT group compared with before training, but not in the COOL group. However, we found neither a significant main effect of group (peak T_rec_
*p* = 0.703, ΔT_rec_
*p* = 0.628) nor an interaction between training period and group (peak T_rec_
*p* = 0.114, ΔT_rec_
*p* = 0.236). The peak T_sk_ during HST did not change after training in either group (*p* = 0.412). Moreover, we found neither a significant main effect of training period (*p* = 0.625) nor an interaction between training period and group (*p* = 0.352). The HR was significantly lower in both groups after training (*p* < 0.001) but we found no significant main effect of group (*p* = 0.996) and no interaction between training period and group (*p* = 0.660).

The HOT group presented declines in the TS after training (*p* = 0.049) and in the RPE (*p* = 0.027) compared with before training, but not in the COOL group. However, there was no significant main effect of group (TS: *p* = 0.852, RPE: *p* = 0.934) and no interaction between training period and group (TS *p* = 0.566, RPE *p* = 0.432). Sweat loss during the HSTs did not change significantly from before to after training in either group (*p* = 0.618); there was no significant main effect of group (*p* = 0.599) and no training period × group interaction (*p* = 0.130). After training, the HOT group showed an increased baseline PV (compared to before training) (*p* = 0.016) but this was not evident in the COOL group. However, no significant main effect of group (*p* = 0.501) and no training period × group interaction (*p* = 0.501) was observed.

### 3.4. Gastrointestinal Damage

Before training, plasma I-FABP levels increased significantly following HST (*p* = 0.042) in COOL group (Pre: 994.9 ± 241.3 pg/mL, Post: 1985.2 ± 777.0 pg/mL), but not HOT group (Pre: 1242.3 ± 748.2 pg/mL, Post: 1867.3 ± 1220.6 pg/mL). However, plasma I-FABP levels did not increased significantly following HST in both COOL (Pre: 1201.1 ± 416.3 pg/mL, Post: 1622.0 ± 594.5 pg/mL) and HOT groups (Pre: 906.9 ± 575.0 pg/mL, Post: 1310.2 ± 852.7 pg/mL) after training. However, we found no significant main effect of training (*p* = 0.213) and no interaction between training period and group (*p* = 0.899).

Figure 2 shows relative changes in plasma I-FABP levels following HST. Plasma I-FABP levels increased significantly after HST (*p* = 0.002) in both HOT and COOL groups before training. However, after training, the exercise-induced plasma I-FABP elevations (Δ) reduced significantly compared with before the training period (*p* = 0.042) in both HOT and COOL groups, with no significant main effect for group (*p* = 0.199) and interaction of training period × group (*p* = 0.388).

### 3.5. Gastric Emptying Rate

The changes in ^13^C-excretion levels are shown in Figure 3. These did not change during the tests in either group (*p* = 0.250) and there was no significant main effect of group (*p* = 0.523) and no interaction between training period and group (*p* = 0.201). As shown in Figure 4, there were no significant differences in T_max_ between the groups before training (*p* = 0.411). We found neither a significant main effect of training (*p* = 0.752) nor an interaction of training × group (*p* = 0.526).

## 4. Discussion

We determined the effects of 10 days of HA on GI damage and the GE rate after a single bout of endurance exercise in a hot environment compared to a cool environment. A novel finding was that 10 days of endurance training in a hot environment improved partially thermoregulation, whereas GI damage and delay of GE rate following endurance exercise in a hot environment were not further attenuated compared with training in a cool environment.

Endurance exercise-induced GI damage is well-established [34,35,36,37,38] but it remains unclear whether endurance training mitigates such damage. In the present study, we found that I-FABP elevations after the HST decreased significantly in both the HOT and COOL groups after 10 days of endurance training, suggesting that short-term endurance training per se attenuates exercise-induced GI damage. It has been previously reported that I-FABP elevation is significantly lower in endurance-trained than in untrained subjects; the former is under less thermoregulatory and cardiovascular strain [37]. As the HR during the HST was significantly lower after endurance training in both groups, cardiovascular function seemed to have potentially improved after training, which may explain the reduction in exercise-induced GI damage. In general, HA increased baseline PV and improves thermoregulation, lowering cardiovascular strain during endurance exercise in a hot environment [18,19]. Thus, we hypothesized that training in a hot environment (i.e., HA training) would attenuate exercise-induced GI damage by maintaining GI blood flow during the HST more so than would training in a cool environment. However, the reductions in exercise-induced GI damage after training did not differ significantly between the HOT and COOL groups. After training, T_rec_ during the HST was lower in the HOT but not the COOL group. In addition, the HOT group had an increased baseline PV (7.7 ± 7.2% increase) after training. It thus appeared that the HOT group achieved HA after 10 days of endurance training in a hot environment. However, that group did not cause further improvements of thermoregulation (i.e., reduction of T_rec_ during the HST), increased baseline PV, or heat tolerance (i.e., an HR reduction during the HST) than the COOL group after training. The extent of HA in the HOT group seems to have been modest compared to previous studies that have reported greater reductions of end-exercise T_rec_ and HR in heat environment [39,40]. Importantly, many previous studies conducted HA training sessions at 38–40 °C [24,41,42], where we did them at 35 °C. The peak T_rec_ during training in the HOT group was 38.30 ± 0.32 °C (CON group 37.96 ± 0.12 °C) in our study, which was lower than in previous studies (T_rec_ > 38.5 °C). Thus, attainment of only partial HA by the HOT group may explain the lack of GI damage attenuation compared to that of the COOL group after training.

The GE rate was not significantly different after HST in either group. After a single exercise session, it can be affected due to changes in gastric motility [43], appetite (which regulates hormonal secretions) [44,45], sympathetic/parasympathetic activities [46], and GI blood flow [44]. On the other hand, any effect of endurance training on exercise-induced GE rate is not well-established. Our findings suggest that short-term, moderate-intensity endurance training does not affect the GE rate after exercise. After endurance training, plasma I-FABP levels were lower in both groups, but the GE rate was not. Exercise-induced plasma I-FABP elevation reflects damage to the small intestine [5], and the GE rate is an indicator of the digestive capacity of stomach fluid [9]. Thus, the differences between these two parameters after exercise are not surprising. A lower GE rate after exercise compromises recovery, as the delivery of macronutrients such glucose or amino acids to working muscles is delayed [9]. Unfortunately, we did not evaluate muscle glycogen or protein resynthesis after exercise. Further study of these parameters, and the rates of appearance of glucose and amino acids in blood after exercise that follows HA, would yield valuable information.

Our work includes several strengths. Many previous studies have investigated the acute effect of exercise on GI function (e.g., I-FABP response). In contrast, the present study revealed the effect of short-term endurance training on GI function under controlled laboratory conditions, which would be the strength of the present study. Moreover, collecting both variables that reflection of absorptive capacity in small intestine (blood I-FABP) and reflection of digestive capacity in stomach (GE rate) as a GI function would be meaningful. On the other hand, the present study has some of limitations. First, the sample size was small (HOT *n* = 8; CON *n* = 7). The present study was conducted over a short period (December to January) to avoid potential effects of seasonal HA. Given the strict limitations in the use of the environmental chamber during the COVID-19 pandemic (the number of subjects in the chamber was minimized to ensure social distancing), we recruited only 15 unacclimated subjects. Second, the heat stress in the HOT group (35 °C) was modest compared to those of previous studies (38–40 °C) [24,41,42]. Thus, evaluation of exercise-induced GI damage and the GE rate after HA at 38–40 °C may yield further information on the impact of HA on GI function.

## 5. Conclusions

Ten days of endurance training in a hot environment improved thermoregulation, whereas GI damage and delay of GE rate following endurance exercise in a hot environment were not attenuated compared with training in a cool environment. Moreover, short-term endurance training did not change GE rate following endurance exercise. Endurance exercise in a hot environment exacerbate GI dysfunction, including increased small intestinal damage, permeability, blood endotoxin and systemic inflammatory cytokine levels [13,14]. Moreover, exercise-induced blood I-FABP elevations delay digests and absorbs nutrients [2]. These responses of GI dysfunction in a hot environment would negatively affect for promoting recovery during post-exercise. In contrast, the present findings suggest that short-term endurance training attenuates GI damage following exercise in a hot environment. Thus, endurance training can be “gut training” in terms of attenuation of GI damage following exercise, which may contribute to promoting recovery during endurance training in a hot environment (e.g., heat acclimation) among endurance athletes. However, addition of heat stress during endurance training does not further reduce the damage.

## Figures and Tables

**Figure 1 nutrients-15-00216-f001:**
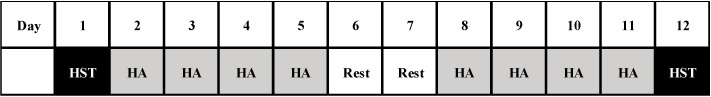
Overview of the study design. HST: heat stress test, HA: heat acclimation.

**Figure 2 nutrients-15-00216-f002:**
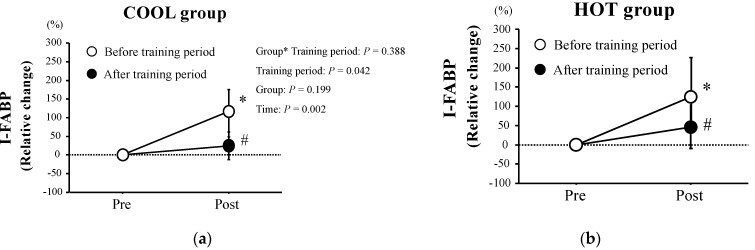
Relative changes in plasma I−FABP concentrations in both COOL (**a**) and HOT groups (**b**). Values are means ± SD. *: Significant difference compared with pre-exercise (Pre). #: Significant difference versus before at the same time point.

**Figure 3 nutrients-15-00216-f003:**
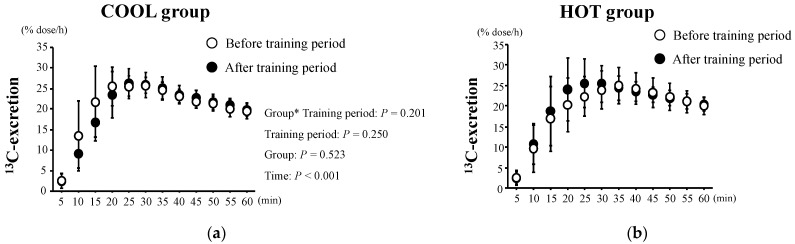
^13^C-excretion after HST before and after the training period in both COOL (**a**) and HOT groups (**b**). Values are means ± SD. *: Significant main effect of training.

**Figure 4 nutrients-15-00216-f004:**
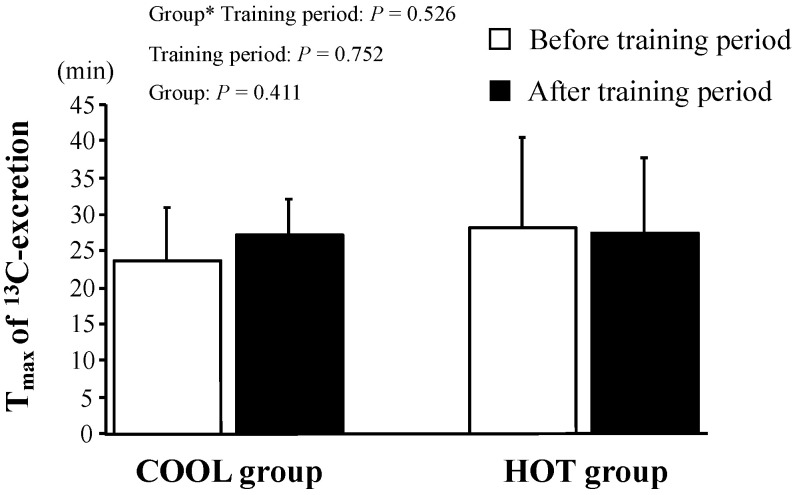
T_max_ of ^13^C-excretion after HST before and after the training period. Values are means ± SD. *: Significant main effect of training.

**Table 1 nutrients-15-00216-t001:** Thermoregulatory and subjective variables during endurance training.

	Day2		Day11	
	HOT Group	COOL Group	HOT Group	COOL Group
Peak T_rec_ (°C)	38.30 ± 0.32 ^†^	37.96 ± 0.12	38.13 ± 0.24 *^,^^†^	37.89 ± 0.22
HR (bpm)	155 ± 16 ^†^	131 ± 7	148 ± 16 *^,^^†^	126 ± 10
Sweat loss (kg)	−0.74 ± 0.4 ^†^	−0.44 ± 0.2	−0.83 ± 0.3 ^†^	−0.30 ± 0.4
TS	7 ± 2 ^†^	3 ± 1	6 ± 1 *^,^^†^	2 ± 1
RPE	6 ± 2 ^†^	4 ± 2	5 ± 1 *^,^^†^	3 ± 1

Values are mean ± SD. *: Significant difference versus Day2. ^†^: Significant difference versus COOL. Variables in table are peak values during 60 min of training session on Day2 and Day11. Trec: rectal temperature. HR: heart rate. TS: thermal sensation. RPE: rate of perceived exertion.

**Table 2 nutrients-15-00216-t002:** Thermoregulatory and subjective variables during HST before and after the training period.

	Before Training Period		After Training Period	
	HOT Group	COOL Group	HOT Group	COOL Group
Rest T_rec_ (°C)	36.94 ± 0.27	36.93 ± 0.32	36.78 ± 0.22	36.87 ± 0.18
Peak T_rec_ (°C)	38.86 ± 0.45	38.65 ± 0.37	38.44 ± 0.42 *	38.50 ± 0.34
ΔT_rec_ (°C)	1.92 ± 0.61	1.71 ± 0.42	1.66 ± 0.42 *	1.63 ± 0.45
T_sk_ (°C)	35.03 ± 0.50	34.62 ± 0.50	35.01 ± 0.52	34.84 ± 1.00
HR (bpm)	178 ± 1	178 ± 5	170 ± 12 *	169 ± 6 *
Sweat loss (kg)	−0.87 ± 0.30	−0.88 ± 0.15	−0.98 ± 0.37	−0.82 ± 0.18
TS	8 ± 2	7 ± 1	7 ± 2 *	7 ± 1
RPE	8 ± 2	7 ± 1	6 ± 2 *	7 ± 2
ΔPV (%)	-	-	7.7 ± 7.2 *	4.6 ± 9.9

Values are mean ± SD. *: Significant difference versus Before. Trec: rectal temperature. Tsk: mean skin temperature. HR: heart rate. TS: thermal sensation. RPE: rate of perceived exertion. PV: plasma volume.

## Data Availability

The data presented in this study are available on request from the corresponding author. The data are not publicly available due to ethical restrictions.

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
