# Peer review of "The Impact of Heat Acclimation on Gastrointestinal Function following Endurance Exercise in a Hot Environment"

_nutrients, 2023, doi:10.3390/nu15010216_

Round 1

Reviewer 1 Report

The manuscript was prepared very well. However, there are some concerns, in part important, so the review articles need revision, see below.

Introduction

·       Include previous history of similar investigations and justify the need for this investigation

Materials and Methods

·       It should include some other data from the sample, such as body composition.

·       What criteria did you use for the selection of participants?

·       Justify sample size

·       Should comply with CARE guidelines (for CAse REports)

Results

·       The results should be presented in clearer tables, redo rows and columns to improve the presentation of results

·       Include the CARE checklist in the development of the study

Discussion

·       You must include references in those paragraphs that do not contain them, so that its content is justified.

·       It should include some hypothesis of the possible mechanisms described from a physiological perspective or include some mechanism of action described.

·       It should include some comparative discussion with other studies related to its purpose, explaining the differences.

·       What does this study contribute? Clarify.

·       Any possible application of the results described?

·       Include a section on strengths and limitations.

Conclusion

·       In the Conclusion section, state the most important outcome of your work. Do not simply summarize the points already made in the body — instead, interpret your findings at a higher level of abstraction. Show whether, or to what extent, you have succeeded in addressing the need stated in the Introduction (or objectives).

Author Response

We thank the reviewers for their valuable comments regarding our manuscript. We have revised the manuscript accordingly. Below is a point-by-point list of the received comments along with our response. In the revised manuscript, we have highlighted the changes using underlined text in red.

Introduction

Comments: Include previous history of similar investigations and justify the need for this investigation

Response: We appreciate this comment. We revised the sentences in introduction.

“Exercise under hot conditions causes greater GI damage than does exercise under thermoneutral conditions [13-15]. Since exercise under hot condition increases core temperature compared with exercise under neutral condition, GI hypoperfusion has been suggested to be exacerbated due to marked increases in skin blood flow and sweat rate to dissipate heat [8,16,17]. On the other hand, several days of endurance training under hot conditions facilitates heat acclimation (HA), leading to increase in baseline plasma volume, lower rectal and skin temperature during endurance exercise under hot conditions [18,19]. These adaptations following HA would cause maintenance of GI blood flow during endurance exercise. Thus, we hypothesized that HA would attenuate exercise-induced GI damage and the delay in the GE rate under hot conditions. However, the effect of HA on exercise-induced GI damage and dysfunction remains unclear. It is necessary to investigate whether exercise-induced blood I-FABP elevation after heat acclimation attenuate or not.” (Page 2, Lines 50-62)

Materials and Methods

Comments: It should include some other data from the sample, such as body composition.

Response: We added another data for body composition (Page 2, Lines 81-83)

Comments: What criteria did you use for the selection of participants?

Response: Inclusion of criteria were healthy active male and non-smoking. We added criteria for the selection of subjects (Page 2, Lines 80-81).

Comments: Justify sample size

Response: We revised the sentences as follows:

“For determination of sample size, we referred to previous studies that demonstrated large effects of thermoregulatory adaptations (e.g., reduction of core temperature during endurance exercise under hot conditions) after several days of HA [24,25].” (Page 2, Lines 77-79)

Comments: Should comply with CARE guidelines (for CAse REports)

Response: We appreciate this comment. We carefully check CARE guidelines.

Results

Comments: The results should be presented in clearer tables, redo rows and columns to improve the presentation of results

Response: We included data for RPE and thermal sensation during HST, baseline plasma volume into table 2.

Comments: Include the CARE checklist in the development of the study

Response: We carefully checked CARE checklist and revised the manuscript.

Discussion

Comments: You must include references in those paragraphs that do not contain them, so that its content is justified.

Response: We added the references into paragraphs that do not contain them. (Page 9, Lines 327, 328, 331)

Comments: It should include some hypothesis of the possible mechanisms described from a physiological perspective or include some mechanism of action described.

Response: We added the sentences of possible mechanism related to our hypothesis in the introduction. (Page 2, Lines 51-58)

Comments: It should include some comparative discussion with other studies related to its purpose, explaining the differences.

Response: In the present study, HOT group failed to attenuate blood I-FABP elevation and delayed GE rate following endurance exercise under hot condition compared with COOL group after training. We included discussion about the lack of GI damage attenuation in the HOT group than in the COOL group after training, with citing previous heat acclimation studies. (Page 8, Lines 312-319)

Comments: What does this study contribute? Clarify.

Response: GI damage and dysfunction delay the appearance of ingested macronutrients (glucose and amino acids) into blood stream, which may impair delivering glucose and/or amino acid to the exercised muscles. Moreover, some of endurance athletes conduct HA training to improve thermoregulatory capacity. Thus, the present study would contribute to developing novel strategy of post-exercise recovery in terms of GI function (digestion and absorption of macronutrients).

Comments: Any possible application of the results described?

Response: We added the possible application as follows:

“Endurance exercise under hot condition exacerbate GI dysfunction, including increased small intestinal damage, permeability, blood endotoxin and systemic inflammatory cytokine levels [13,14]. Moreover, exercise-induced blood I-FABP elevations delay digests and absorbs nutrients [2]. These responses of GI dysfunction under hot condition would negatively affect for promoting recovery during post-exercise. In contrast, the present findings suggest that short-term endurance training attenuates GI damage following exercise under hot condition. Thus, endurance training can be “gut training” in terms of attenuation of GI damage following exercise, which may contribute to promoting recovery during endurance training under hot condition (e.g., heat acclimation) among endurance athletes. However, addition of heat stress during endurance training does not further reduce the damage.” (Page 9, Lines 354-365)

Comments: Include a section on strengths and limitations.

Response: We added section of strength and limitations. (Page 9, Lines 335-349).

Conclusion

Comments: In the Conclusion section, state the most important outcome of your work. Do not simply summarize the points already made in the body — instead, interpret your findings at a higher level of abstraction. Show whether, or to what extent, you have succeeded in addressing the need stated in the Introduction (or objectives).

Response: We appreciate this comment. We revised the conclusion as follows:

“Ten days of endurance training under hot condition improved thermoregulation, whereas GI damage and delay of GE rate following endurance exercise under hot con-dition were not attenuated compared with training under cool condition. Moreover, short-term endurance training did not change GE rate following endurance exercise. Endurance exercise under hot condition exacerbate GI dysfunction, including increased small intestinal damage, permeability, blood endotoxin and systemic inflammatory cytokine levels [13,14]. Moreover, exercise-induced blood I-FABP elevations delay digests and absorbs nutrients [2]. These responses of GI dysfunction under hot condition would negatively affect for promoting recovery during post-exercise. In contrast, the present findings suggest that short-term endurance training attenuates GI damage following exercise under hot condition. Thus, endurance training can be “gut training” in terms of attenuation of GI damage following exercise, which may contribute to promoting recovery during endurance training under hot condition (e.g., heat acclimation) among endurance athletes. However, addition of heat stress during endurance training does not further reduce the damage.” (Page 9, Lines 351-365)

Reviewer 2 Report

Comments to the Authors of manuscript number: nutrients-2105234 entitled “The impact of heat acclimation on gastrointestinal function following endurance exercise under hot condition”.

The study was performed on athletes subjected to the two types of exercises. But, I cannot see the real analysis of thermoregulation.

1. L 13, 65- please avoid to use “we”

2. L 17,18 - HST should be explained

3. L 19 – plasma intestinal?

4. L 23 – abbr. should be explained

5. L 36 – what is plasma intestinal FA?

6. L 37 why not D-lactate?

7. L 66 – the hypothesis should be given earlier

8. L 97 – rectal temperature is not the core body temperature. It is considered internal body temperature from the medical point of view.

9. L 98 – what is mean skin temperature? It is a mean value of a few measurements of the same point at different time, or the mean of many measurements of various areas of the skin?

10. L 100 – the reference for such control of sweat loss should be given.

11. L 114- kp?

12. L 147 – a repetition of L 142

13. L 149 – the range of detection?

14. why was the period of 10 days? It should be explained.

15. L 262 – please explain how thermoregulation was improved

16. plasma volume changes when urine is produced not only sweat. Was the volume of urine detected? What about saliva production?

17. L 174 – expansion?

18. L 274-275, L 284- it is not evident

19. was thermal analysis of body performed? It could show the influence of the HA on the thermal core body

20. Was the skin blood flow determined?

21. L 319 what parameters exactly can indicate it?

Author Response

We thank the reviewers for their valuable comments regarding our manuscript. We have revised the manuscript accordingly. Below is a point-by-point list of the received comments along with our response. In the revised manuscript, we have highlighted the changes using underlined text in red.

Comments: L 13, 65- please avoid to use “we”

Response: We deleted the words of “we”. (Page 1, Line 13) (Page 2, Line 69)

Comments: L 17,18 - HST should be explained

Response: We revised the sentences. (Page 1, Lines 17-20)

Comments: L 19 – plasma intestinal?

Response: In the many previous studies, blood intestinal fatty acid-binding protein (I-FABP) was analyzed using plasma samples (Snipe et al 2018a; Snipe et al. 2018b; Hill et al. 2020). Therefore, we described “plasma I-FABP levels”.

Comments: L 23 – abbr. should be explained

Response: Abbreviations for I-FABP was described. (Page 1, Lines 18-19)

Comments: L 36 – what is plasma intestinal FA?

Response: Plasma intestinal fatty acid binding protein (I-FABP) is an indirect marker of damage to the small intestinal. (Page 1, Lines 36-37)

Comments: L 37 why not D-lactate?

Response: In the present study, we focused on small intestinal damage following exercise. Exercise-induced plasma I-FABP elevations have been utilized as an indirect marker of damage to the small intestine (Chantler et al. 2021). Therefore, we selected plasma I-FABP levels in the present study as an indication of small intestinal damage.

Comments: L 66 – the hypothesis should be given earlier

Response: We changed the place of the sentence of hypothesis into introduction section (Page 2, Lines 58-60).

Comments: L 97 – rectal temperature is not the core body temperature. It is considered internal body temperature from the medical point of view.

Response: We appreciate this comment. We reconsidered rectal temperature as an internal body temperature.

Comments: L 98 – what is mean skin temperature? It is a mean value of a few measurements of the same point at different time, or the mean of many measurements of various areas of the skin?

Response: Mean skin temperature have been used in many previous studies to evaluate the effect of heat acclimation on thermoregulatory function (Lorenzo et al. 2010; Neal et al. 2016; Travers et al. 2020). It was calculated using the following equation (Ramanathan et al. 1964): Mean skin temperature = 0.3 × (chest + arm) + 0.2 × (thigh + calf).

Mean skin temperature was measured halfway between the nipple and clavicle for the chest, at 60% of the distance between the acromion and elbow for the arm, halfway between the greater trochanter and knee for the thigh, and at 30% of the distance between the knee and lateral malleolus for the calf. (Page 3, Lines 136-142)

Comments: L 100 – the reference for such control of sweat loss should be given.

Response: We added the reference. (Page 3, Line 111)

Comments: L 114- kp?

Response: The kp is abbreviation of kilopond, and it is pedaling workload. We added abbreviation of kp. (Page 3, Lines 127-128)

Comments: L 147 – a repetition of L 142

Response: We deleted the repetition sentence.

Comments: L 149 – the range of detection?

Response: Coefficient of variation (CV) also known as relative standard deviation (RSD) and it is defined as the ratio of the standard deviation to the mean.

Comments: why was the period of 10 days? It should be explained.

Response: In the present study, all subjects performed 8 training sessions and rested 2 days (total 10 days of training period). Therefore, we described 10 days of training period.

Comments: L 262 – please explain how thermoregulation was improved

Response: Several physiological factors contribute to improvement of thermoregulation during exercise after heat acclimation, including increased baseline plasma volume, sweat rate and greater heat dissipation from skin. Indeed, many previous studies established that endurance training under hot condition (heat acclimation) improved thermoregulation (e.g., lower rectal temperature) during exercise under hot condition due to increased baseline plasma volume, increased sweat rate and skin blood flow during exercise. In the present study, HOT group induced increase in baseline plasma volume after the training. We think that increase in baseline plasma volume would be a factor for improved thermoregulation (i.e., lower Trec) during HST in the HOT group after the training.

Comments: plasma volume changes when urine is produced not only sweat. Was the volume of urine detected? What about saliva production?

Response: Unfortunately, we did not measure urine and saliva volume in the present study. However, calculation of plasma volume using blood sample have been well established (Dill and Costill. 1974), and it was already utilized to determine plasma volume shift (Lorenzo et al. 2010; Snipe et al. 2018b; Sumi et al. 2022). Thus, we think that changes in plasma volume calculated by blood sample would not be problem.

Comments: L 174 – expansion?

Response: We replaced “plasma volume expansion” with “increase in baseline plasma volume” throughout in the revised manuscript.

Comments: L 274-275, L 284- it is not evident

Response: We revised the sentences. (Page 8, Lines 300, 310-311)

Comments: was thermal analysis of body performed? It could show the influence of the HA on the thermal core body

Response: We did not perform thermal analysis of body. However, previous studies that investigated the effect of heat acclimation on thermoregulation during HST selected rectal temperature to evaluate thermoregulation before and after heat acclimation (Lorenzo et al. 2010; Neal et al. 2016; Travers et al. 2020). Therefore, we believe that the determination of rectal temperature during exercise before and after heat acclimation was an appropriate procedure to evaluate thermoregulatory function during exercise.

Comments: Was the skin blood flow determined?

Response: Unfortunately, we did not measure skin blood flow during HST.

Comments: L 319 what parameters exactly can indicate it?

Response: In the present study, “improved thermoregulation” reflects lower rectal temperature during HST after training. Because many previous studies used rectal temperature to determine thermoregulation during exercise before and after heat acclimation (Lorenzo et al. 2010; Neal et al. 2016; Travers et al. 2020), we measured rectal temperature as an indication of thermoregulation during exercise in the present study.

Round 2

Reviewer 1 Report

The authors have highlighted all the suggestions and the manuscript has gained in quality.

Congratulations

Author Response

We thank the reviewer for your valuable comments regarding our manuscript.

Reviewer 2 Report

Thank Authors for the explanation. I have no comments.

Author Response

(The authors gave the same response as above.)
